# Changes in blood pressure thresholds for initiating antihypertensive medication in patients with diabetes: a repeated cross-sectional study focusing on the impact of age and frailty

Martina Ambrož [ID],[1] Sieta T de Vries [ID],[1] Grigory Sidorenkov [ID],[1] Klaas Hoogenberg,[2] Petra Denig [ID] [1]

¹University of Groningen, University Medical Centre Groningen, Groningen, The Netherlands
²Martini Ziekenhuis, Groningen, The Netherlands

**Correspondence to**
Martina Ambrož;
m.ambroz@umcg.nl

## ABSTRACT

**Objective** To assess trends in systolic blood pressure (SBP) thresholds at initiation of antihypertensive treatment in patients with type 2 diabetes and the impact of age and frailty on these trends.

**Study design and setting** A repeated cross-sectional cohort study (2007–2014) using the Groningen Initiative to Analyse Type 2 diabetes Treatment database was conducted. The influence of calendar year, age or frailty and the interaction between year and age or frailty on SBP thresholds were assessed using multilevel regression analyses adjusted for potential confounders.

**Results** We included 4819 patients. The mean SBP at treatment initiation was 157 mm Hg in 2007, rising to 158 mm Hg in 2009 and decreasing to 151 mm Hg in 2014. This quadratic trend was significant (p<0.001). Older patients initiated treatment at higher SBP, but similar decreasing trends after 2009 were observed in all age groups. There were no significant differences in SBP thresholds between patients with different frailty groups. The association between year and SBP threshold was not influenced by age or frailty.

**Conclusion** After an initial rise, the observed SBP thresholds decreased over time and were not influenced by age or frailty. This is in contrast with changed guideline recommendations towards more personalised treatment during the study period and illustrates that changing prescribing practice may take considerable time. Patient-specific algorithms and tools focusing on when and when not to initiate treatment could be helpful to support personalised diabetes care.

## INTRODUCTION

Treatment of hypertension in patients with type 2 diabetes (T2D) reduces cardiovascular risk, but guideline recommendations on when to initiate antihypertensive treatment to best balance the benefits and risks of treatment have changed over time. In the past, the recommended systolic blood pressure (SBP) threshold for treatment initiation ranged from 130 mm Hg to 140 mm Hg.[1–7]

> ### Strengths and limitations of this study
>
> ► This is a first study investigating trends in systolic blood pressure thresholds at initiation of antihypertensive treatment in patients with type 2 diabetes.
> ► Real-world data from primary care of a large number of patients with type 2 diabetes initiating antihypertensive treatment were used.
> ► The analyses were adjusted for a range of possible confounders and multiple imputation was used to reduce any possible bias that could have occurred due to missing data.
> ► The number of included general practices and patients fluctuated over the years.
> ► Frailty might have been underestimated due to incomplete recording of International Classification of Primary Care diagnoses in medical records.

A personalised approach, however, has been advocated in the last decade for older and/or frail patients, who are at increased risk of adverse outcomes related to low blood pressure (BP) levels.[4 6 7] Since 2011, treatment guidelines started to recommend higher SBP thresholds in these patients (figure 1).[3–11] A recent interview study showed that Dutch general practitioners were indeed somewhat reluctant to initiate antihypertensive treatment in older and/or frail patients.[12] Studies showing at which SBP thresholds physicians initiate antihypertensive treatment in older or frail patients are lacking.

A Danish study observed an average SBP level in the general population of 148 mm Hg before they received antihypertensive treatment in the period from 1976 to 2004.[13] Trend studies on antihypertensive medication use and hypertension control in individuals with T2D show that the percentage of people achieving the recommended

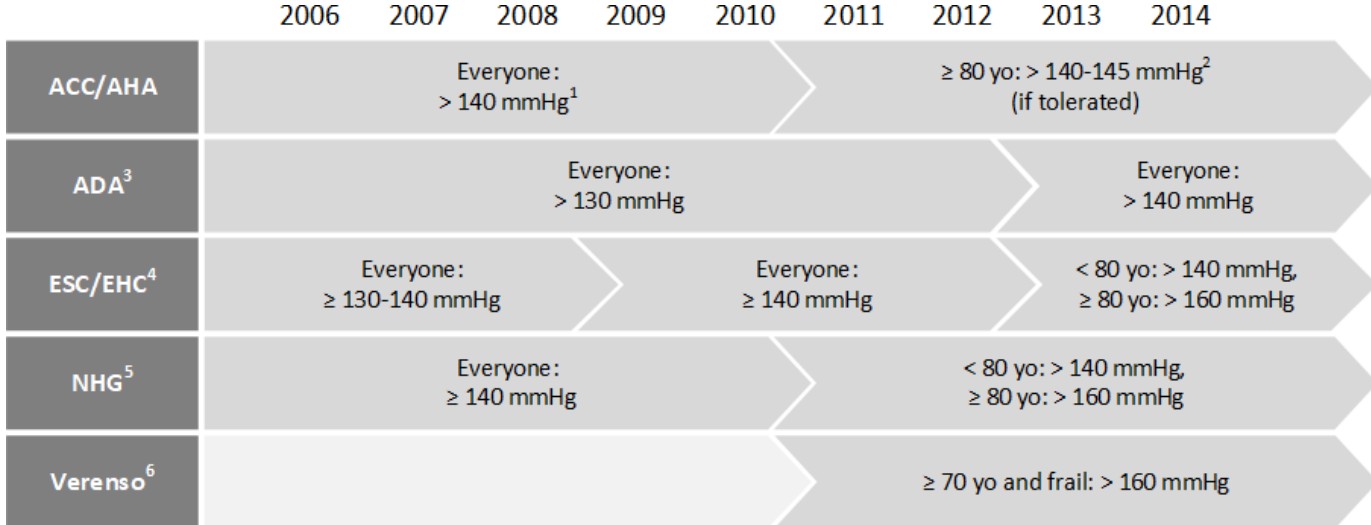

**Figure 1** American, European and Dutch guideline recommended systolic blood pressure values for initiation of antihypertensive treatment in patients with type 2 diabetes over the years (1=American College of Cardiology/American Heart Association (ACC/AHA) guidelines[8]; 2=ACC/AHA Expert Consensus Document on Hypertension in the Elderly[4]; 3=American Diabetes Association (ADA) Standards of Medical Care in Diabetes[3 9]; 4=European Society of Cardiology/Euro Heart Care (ESC/EHC) guidelines for hypertension management[5 6 10]; 5=Dutch College of General Practitioners (NHG) cardiovascular risk management guidelines[1 11]; 6=Verenso multidisciplinary guidelines for the management of diabetes[7]). yo = years old

SBP target of <140 mm Hg increased over the last 20 years.[14 15] An observational study conducted in the Netherlands showed that the mean achieved SBP decreased from 155 mm Hg in 1998 to 140 mm Hg in 2008 in all age groups, with a mean SBP being lower in younger patients. No relevant differences in trends were observed between different age groups.[16] This indicates that BP control in patients with T2D has generally improved over time. The extent to which the more recent personalised guideline recommendations are followed, however, is not clear. This may depend on physician or practice characteristics,[17 18] resulting in variability between treatment decisions.[19]

The aim of this study was to assess trends in SBP thresholds for initiating antihypertensive medication in patients with T2D and the impact of changed treatment recommendations for older and frail patients. We looked at the period between 2007 and 2014, for which we hypothesised that SBP thresholds would remain similar among young and non-frail patients but would increase among older and/or frail patients. Our secondary aim was to assess to what extent SBP thresholds for treatment initiation varied across general practices.

## METHODS
### Study design and population
This was a repeated cross-sectional dynamic cohort study for the years 2007–2014. The Groningen Initiative to Analyse Type 2 diabetes Treatment (www.giantt.nl) database was used, which contains anonymous electronic medical records data of patients with T2D treated in primary care in the north of the Netherlands.

Patients were included per calendar year when they had a diagnosis of T2D and were ≥18 years. We excluded patients who were not included in the database for at least 365 days before antihypertensive treatment initiation and did not initiate treatment with an antihypertensive (Anatomical Therapeutic Chemical codes C03, C04, C07, C08, C09) in that year. Antihypertensive treatment initiation was defined as an initial prescription without a prescription of any antihypertensive drug in the preceding 365 days. Furthermore, patients were excluded when they did not have a documented SBP level within 120 days before or at the day of treatment initiation or when they initiated treatment with three or more drug classes, since it is unlikely that this was a true initiation. It is assumed that these are patients with prevalent antihypertensive treatment, who entered the dynamic cohort during the study period. Moreover, we excluded patients initiating propranolol or a loop diuretic (furosemide, bumetanide), since these are commonly prescribed for other indications (ie, migraine prophylaxis or short-term use in patients with oedema, respectively). No approval from an ethics committee is needed for studies using data from anonymous medical records in the Netherlands. An exemption letter from University Medical Center Groningen Medical Ethics Review Board was obtained (reference number M19.235285).

### Outcome variable
The outcome was the patient's most recent office SBP level in the 120 days before or on the day of antihypertensive treatment initiation.

### Explanatory variables
The following explanatory variables were included: calendar year, age or frailty of the patient and the interaction between year and age or frailty. Age was calculated

on 1 January of each year and was categorised in four groups (<60 years, 60–69 years, 70–79 years and ≥80 years) related to cut-off values mentioned in several guidelines.[7 11 20] Frailty was calculated using a previously developed electronic Frailty Index (eFI), which is based on 140 International Classification of Primary Care (ICPC) coded symptoms and diagnoses from the medical history as well as the existence of polypharmacy.[21] These ICPC codes are grouped into 36 deficits, for which patients get points. For chronic conditions, a diagnostic code anytime in the past is included, whereas for short-term or episodic conditions only diagnostic codes from the past year are included. The sum of the points from the deficits divided by 36 is the indication of frailty and can take a value between 0 (patient has no deficits) and 1 (patient has all possible deficits). Since all included patients had diabetes, we excluded diabetes from the eFI, thus focusing on additional frailty. There are no validated clinical cut-off values for the eFI; therefore, we categorised the scores in tertiles based on the eFI values in our study population to compare less, medium and more frail patients.

## Confounders

The following patient characteristics were included as possible confounders: sex, diabetes duration (<2 years or ≥2 years), presence of dyslipidaemia (low density cholesterol (LDL) ≥2.5 mmol/L), glycated haemoglobin level (HbA$_{1c}$ <7% or ≥7%), estimated glomerular filtration rate (eGFR; ≤60 mL/min or >60 mL/min), presence of elevated albuminuria (albumin creatinine ratio ≥30 mg/g or albumin in 24 hours urine ≥300 mg), history of cardiovascular events (presence yes/no of myocardial disease, heart failure or stroke), body mass index (BMI; <24.9 kg/m², 25–29.9 kg/m² or ≥30 kg/m²), number of prescribed chronic medication (continuous variable), number and type of glucose lowering treatment (none, one oral, two oral, three or more oral and/or insulin) and lipid lowering treatment (none or one/more drug classes). The most recent laboratory values available in the 365 days before or 7 days after treatment initiation were used. BMI was calculated from weight and height or extracted from the database in case these were not available. The eGFR was calculated from serum creatinine using the Modification of Diet in Renal Disease-4 equation for the years 2007–2009, and using the Chronic Kidney Disease Epidemiology Collaboration equation from 2010 onwards, since the standard way of calculating eGFR in the Netherlands changed during the study period.[22] In case serum creatinine was not available, the eGFR was extracted from the database when available. Prescribed medication was assessed in the 120 days before or at the day of treatment initiation.

## Missing data

There were no missing data for the explanatory variables. Values of confounders with <20% of missing values were imputed using multiple imputation by chained equation.[23] For patients without albuminuria measurements (47%), we assumed that they did not have elevated albuminuria, since urine samples were less likely to be collected in our study period for patients without suspected renal function problems.

## Analyses

Descriptive statistics were performed to examine patient characteristics per calendar year.

We conducted multilevel regression analyses with a two-level random intercept model to account for patients being nested within general practices. First, using the empty model, which includes only the outcome variable, we calculated the intraclass correlation coefficient to assess the variance that is attributed to general practices. Second, we added the confounders to assess the overall trend over the years. We compared a linear and a quadratic model using the Wald test to choose the best fitting final model. In the final model, we assessed the effect of age or frailty on the trends by adding the explanatory variables and the interaction between year and age or frailty on SBP levels at treatment initiation.

Additional subgroup analyses were conducted for each age and frailty group to assess changes over time in these subpopulations using the final model. After applying Bonferroni correction for multiple testing, significance levels were set at p<0.0125 (per age group) and p<0.0167 (per frailty group).

Furthermore, we conducted sensitivity analyses using the average of the last two SBP levels instead of a single SBP measurement and using eFI as a continuous variable in the final model.

The analyses were conducted in Stata V.14 (Stata Corp.).

## Patient and public involvement

Patients and public were not involved in this study.

## RESULTS

A total of 4819 patients initiating antihypertensive treatment in the period 2007–2014 were included (table 1). A flow chart of excluded patients per calendar year is presented in online supplementary figure 1. Patient characteristics were generally similar throughout the years (online supplementary table 1). Seventy-four per cent of included patients had no missing values.

## Trends in SBP thresholds

The mean SBP level at antihypertensive treatment initiation significantly changed over time from 157 mm Hg (SD 22 mm Hg) in 2007, rising to 158 mm Hg (SD 21 mm Hg) in 2009 and thereafter decreasing to 151 mm Hg (SD 22 mm Hg) in 2014 (figure 2A). This quadratic trend was statistically significant (p<0.001).

## Age and frailty

Older patients initiated treatment at significantly higher SBP thresholds than younger patients but age did not significantly influence the relationship between calendar year and SBP threshold (table 2, figure 2B). In the analyses

**Table 1** Characteristics of included patients (n=4819)

| | Included patients |
|---|---|
| **Calendar year** | |
| 2007 | 328 |
| 2008 | 423 |
| 2009 | 564 |
| 2010 | 591 |
| 2011 | 811 |
| 2012 | 735 |
| 2013 | 718 |
| 2014 | 649 |
| **Females; N (%)** | 2259 (47) |
| **Age in years; N (%)** | |
| <60 | 1620 (34) |
| 60–69 | 1585 (33) |
| 70–79 | 1068 (22) |
| ≥80 | 546 (11) |
| **Frailty in electronic Frailty Index score; N (%)** | |
| Less frail: 0–0.03 | 2070 (43) |
| Medium frail: 0.06–0.09 | 1628 (34) |
| More frail: 0.11–0.40 | 1121 (23) |
| **Systolic BP at initiation in mm Hg; mean±SD** | 155±22 |
| **Diastolic BP at initiation in mm Hg; mean±SD** | 85±12 |
| **Diabetes duration <2 years; N (%)** | 1259 (26) |
| **HbA$_{1c}$<7%; N (%)*** | 2717 (61) |
| **BMI in kg/m$^2$; N (%)†** | |
| <24.9 | 751 (17) |
| 25–29.9 | 1788 (41) |
| ≥30 | 1853 (42) |
| **Dyslipidaemia; N (%)‡** | 2263 (57) |
| **eGFR ≤60 mL/min/1.73m$^2$; N (%)§** | 542 (13) |
| **Elevated albuminuria (%)¶** | 101 (4) |
| **History of cardiovascular disease; N (%)** | |
| Myocardial disease** | 263 (5) |
| Heart failure†† | 90 (2) |
| Stroke‡‡ | 212 (4) |
| **Number of chronic medication at initiation; mean±SD** | 3.6±2.5 |
| **Glucose lowering medication at initiation; N (%)** | |
| No medication | 1269 (26) |
| 1 oral | 1937 (40) |
| 2 oral | 982 (21) |
| 3 oral or more and/or insulin | 631 (13) |

Continued

**Table 1** Continued

| | Included patients |
|---|---|
| **Treated with a lipid lowering drug; N (%)** | 2749 (57) |
| **Initiated drug class; N (%)** | |
| Renin–angiotensin–aldosterone system inhibitor | 2645 (55) |
| Diuretic | 762 (16) |
| Beta blocker | 689 (14) |
| Calcium channel blocker | 240 (5) |
| Combination of antihypertensives | 474 (10) |

*Haemoglobin A1C (HbA$_{1c}$): 352 (7.3 %) missing values.
†Body mass index (BMI): 427 (8.9 %) missing values.
‡LDL cholesterol: 874 (18.1%) missing values.
§Estimated glomerular filtration rate (eGFR): 677 (14.0 %) missing values.
¶Albuminuria: 2274 (47.2%) missing values.
**Acute myocardial infarction (ICPC code K75) in the last year or other/chronic ischaemic heart disease (ICPC code K76) anytime in history.
††Heart failure (ICPC code K77) anytime in history.
‡‡Transient cerebral ischaemia (ICPC code K89) in the last year or stroke/cerebrovascular incident (ICPC code K90) anytime in history.
BP, blood pressure; ICPC, International Classification of Primary Care.

per age group, the SBP level at initiation changed significantly (quadratic model) over the years in patients aged between 60 and 69 years (p=0.001).

Frailty did not influence SBP thresholds for treatment initiation and it did not significantly influence the relationship between calendar year and SBP threshold (table 2, figure 2C). In the analyses per frailty group, the SBP level at initiation changed significantly (quadratic model) over the years in the less frail (eFI 0–0.03; p<0.001) and more frail (eFI 0.11–0.40; p=0.001) patients.

The sensitivity analyses using the mean of the last two SBP levels (online supplementary figure 2A-C and online supplementary table 2) and using eFI as a continuous variable (online supplementary table 3) showed similar results.

### Variation between general practices

Of the total variation in SBP level at antihypertensive treatment initiation, 3.2% could be explained by differences between general practices (table 2, intraclass correlation coefficient=0.032).

## DISCUSSION
### Summary

This study shows that, after an initial rise up to 2009, SBP thresholds for antihypertensive treatment initiation decreased over time in a large cohort of patients with T2D treated in primary care. This trend occurred regardless of age and frailty, which was in contrast to our hypothesis

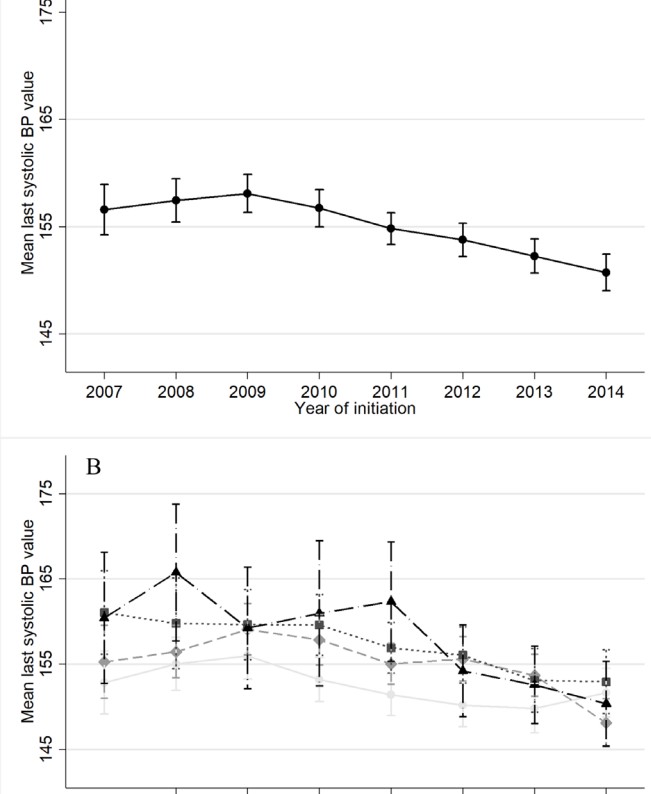

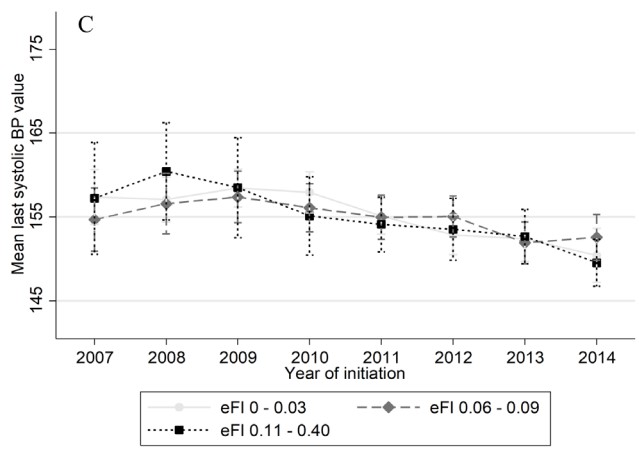

**Figure 2** Mean last systolic blood pressure (BP) value with 95% CIs before/at antihypertensive treatment initiation (A) through the years; (B) through the years in different age groups; (C) through the years in different frailty groups. eFI, electronic Frailty Index.

given the changes in guideline recommendations. The variation in SBP thresholds for treatment initiation that could be attributed to general practices was small.

### Strengths and limitations

The strength of our study is the large number of patients included using real-world data. Furthermore, it is a first study investigating trends in SBP thresholds at initiation of antihypertensive treatment. We focused on the period from 2007 to 2014, expecting that the shift towards more personalised diabetes care around 2011 could be observed during this period. It is possible that changes occurred in more recent years. The first limitation is the fluctuating number of general practices over the years, with different numbers of patients and practices included in each year cohort. Since little variation was explained at practice level, it is unlikely that this affected our primary findings. Second, there were some demographic differences between the patients with and without missing data (online supplementary table 4). This bias was reduced by multiple imputation.[24] Third, we could not include smoking or date of hypertension diagnosis as confounders due to amount and variability of missing values over the years. Furthermore, ICPC codes do not provide information about the severity of the comorbidities. Therefore, there may be some residual confounding which was not accounted for. Fourth, incomplete coding of ICPC diagnoses in electronic medical records may result in underestimation of frailty. Finally, SBP levels show intraindividual variability and may include higher values caused by 'white coat' hypertension.[25] However, analysis of the mean last two SBP levels did not change our main findings.

### Comparison with existing literature

The observed trends are not in line with changes in treatment guidelines, where higher thresholds were recommended in the older and/or frail patients, particularly in the later years. Several reasons can explain this discrepancy between our hypothesis and findings. First, Dutch healthcare practitioners may have felt pressured to initiate antihypertensive treatment at lower SBP levels in all patients with diabetes after the introduction of performance indicators in 2007.[26] From 2008 onwards, they received yearly feedback on the percentage of patients achieving SBP levels of <140 mm Hg in their own practice as compared with other practices in the region. Although a previous study in this population did not show an increase in overtreatment after the introduction of performance indicators,[27] concerns about the negative impact of such measures have been raised.[28] In addition, nurse practitioners increasingly became the pivot of diabetes care[26] and their educational material recommended a unified SBP target of 140 mm Hg at least until the end of our study period.[29] It is also possible that the practitioners did not adhere to treatment guidelines either due to lack of familiarity, understanding or agreement with them.[30–32] An interview study conducted in 2015 and 2016, however, suggests that Dutch general practitioners did support the idea of using a higher threshold for initiation of antihypertensive treatment in older and frail patients.[12] Nevertheless, we demonstrated that guideline changes were not yet implemented more than 3 years after being published.

**Table 2** Influence of calendar year and age or frailty on blood pressure thresholds (multilevel analysis)

| | β | 95% CI | P value | |
|---|---|---|---|---|
| **Age*** | | | | |
| Calendar year | −0.107 | −1.429 to 1.215 | 0.874 | <0.001† |
| Calendar year$^2$ | −0.111 | −0.248 to 0.027 | 0.114 | |
| Age <60 years | −8.066 | −10.411 to −5.723 | <0.001 | |
| Age 60–69 years | −4.115 | −6.369 to −1.861 | <0.001 | |
| Age 70–79 years | −1.168 | −3.407 to 1.072 | 0.307 | |
| Age ≥80 years | Reference group | | | |
| Interaction year* age | None are significant | | | |
| **Frailty‡** | | | | |
| Calendar year | 0.247 | −1.100 to 1.593 | 0.719 | <0.001† |
| Calendar year$^2$ | −0.159 | −0.299 to 0.018 | 0.027 | |
| Frailty 0–0.03 | −0.060 | −1.734 to 1.614 | 0.944 | |
| Frailty 0.06–0.09 | 0.127 | −1.519 to 1.772 | 0.880 | |
| Frailty 0.11–0.40 | Reference group | | | |
| Interaction year* frailty | None are significant | | | |

The intraclass correlation coefficient calculated from the empty model was 0.032.
*The age model was adjusted for sex, duration of diabetes, number of chronic medication at initiation, number and/or type of glucose lowering therapy, lipid lowering therapy, presence of albuminuria, presence of dyslipidaemia, haemoglobin A1C (HbA$_{1c}$), history of cardiovascular events, estimated glomerular filtration rate and body mass index.
†Joined significance of calendar year and calendar year$^2$ using the Wald test.
‡The frailty model was adjusted for sex, duration of diabetes, number and/or type of glucose lowering therapy, lipid lowering therapy and HbA$_{1c}$.

Antihypertensive treatment was initiated at relatively high SBP levels in patients of all ages but started to decline after 2009, suggesting that it took many years before the United Kingdom Prospective Diabetes Study recommended threshold of 140 mm Hg for treatment initiation was implemented in practice.[2] Other studies showed that the percentage of patients achieving an SBP <140 mm Hg increased in the last two decades but that this increase was smaller in older patients.[15] A recent US study showed that the trends in SBP levels in patients ≥60 years decreased until 2010 and remained relatively stable in the 6 years thereafter.[33] Although this study was not restricted to patients with T2D, this seems in contrast with our findings where SBP levels after 2009 decreased in all age groups.

To our surprise, frailty did not influence the SBP threshold for treatment initiation. The frailty range in our study was rather low, which could indicate that antihypertensive treatment is initiated when the patients are still relatively fit. On the other hand, frailty can easily be overlooked due to subtle manifestations, lack of time or a lack of consensus on the best way to assess it.[34] Although the eFI was previously able to identify frailty comparable to the Groningen Frailty Index,[21] it might not be in line with the practitioner's perception of a patient's frailty. Therefore, we conducted a post-hoc analysis using the number of chronic medication a patient was receiving at initiation as a proxy for frailty. We observed that the patients being treated with more than three chronic medication (median) initiated treatment at lower SBP levels than those being treated with three or less (online supplementary figure 3 and online supplementary table 5). This finding suggests that lower instead of higher thresholds are used for frail patients.

Only a small part of the variation in our study could be attributed to differences between practices. This suggests that patient characteristics determine the threshold to a greater extent than practice characteristics. We only looked at variation between practices which may include decisions of two or more general practitioners within one practice. Unfortunately, we could not conduct analyses at the level of individual practitioners.

## CONCLUSION AND IMPLICATIONS

The observed SBP thresholds at initiation of antihypertensive treatment decreased after 2009. This trend was not influenced by age or frailty, which is in contrast with changes in treatment recommendations, and may be explained by the introduction of performance indicators. Our study illustrates that changing prescribing practice may take considerable time and only publishing new recommendations might not be sufficient for their successful implementation. On one hand, patient-specific algorithms and tools to support the timely start of antihypertensive treatment in younger patients are needed. On the other hand, also algorithms and tools to prevent the initiation of too early or strict

antihypertensive treatment in older and frail patients should be developed. Furthermore, performance indicators should include the aspect of more personalised treatment recommendations. Further research is needed to assess the underlying reasons and extent of the delay in the implementation of personalised diabetes care and evaluate the impact of strategies to speed up the uptake of recommendations.

**Contributors** MA contributed to the development and formulation of the research question, conducted the analysis, contributed to the interpretation of data, wrote the manuscript, reviewed and edited the manuscript, and is the guarantor of this work and, as such, takes responsibility for the integrity of the data and the accuracy of the data analysis. STdV contributed to the development and formulation of the research question, conducted the analysis, contributed to the interpretation of data and reviewed and edited the manuscript. GS contributed to the development of the analysis, the interpretation of data and reviewed and edited the manuscript. KH contributed to the development and formulation of the research question, the interpretation of data and reviewed and edited the manuscript. PD contributed to the development and formulation of the research question, development of the analysis, the interpretation of data and reviewed and edited the manuscript.

**Funding** This project has received funding from the European Union's Horizon 2020 research and innovation programme under the Marie Skłodowska-Curie grant agreement No 754425.

**Disclaimer** The funding source had no involvement in the study.

**Competing interests** None declared.

**Patient and public involvement** Patients and/or the public were not involved in the design, or conduct, or reporting, or dissemination plans of this research.

**Patient consent for publication** Not required.

**Provenance and peer review** Not commissioned; externally peer reviewed.

**Data availability statement** Data are available upon reasonable request.

**ORCID iDs**
Martina Ambrož http://orcid.org/0000-0003-1319-4898
Sieta T de Vries http://orcid.org/0000-0001-6090-2434
Grigory Sidorenkov http://orcid.org/0000-0002-1926-8862
Petra Denig http://orcid.org/0000-0002-7929-4739

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
