## [Reviewer comments · BMJ Open]

ARTICLE DETAILS

TITLE (PROVISIONAL)	Changes in blood pressure thresholds for initiating antihypertensive medication in diabetes patients: A repeated cross-sectional study focusing on the impact of age and frailty
AUTHORS	Ambrož, Martina; de Vries, Sieta; Sidorenkov, Grigory; Hoogenberg, Klaas; Denig, Petra

VERSION 1 – REVIEW

REVIEWER	Gregory Wozniak American Medical Association United States
REVIEW RETURNED	2-Apr-2020

GENERAL COMMENTS	Does the data include other comorbidities that could be included in the model? The authors use the number of chronic condition medications as a control, but there may be chronic conditions where no medications are prescribed. Also, the severity of those conditions and the date of hypertension diagnosis should be accounted for. Duration of diabetes is accounted for. Or are these accounted for in the frailty index. A brief description of the frailty index would be helpful for readers who are unfamiliar with the index. The frailty index has a range of 0-1, and was categorized in to tertiles. The tables have the ranges of 0 – 0.06, 0.08 – 0.11 and 0.14-.036. I don't see where the authors clarify only presenting those ranges, and how what is presented corresponds to tertiles of the index. Regarding references, just one suggestion. There are more recent guidelines for management of blood cholesterol which the authors should review, ACC/AHA 2018 Guideline on the Management of Blood Cholesterol. Comments on sensitivity analysis: The authors examined the use of the mean of SBP. What about sensitivity of results where rather than using SBP within 120 days of medication initiation, some shorter look back, eg, 30, 60 or 90.? Shorter look backs may reduce the sample, but 120 days would be from a fairly old visit, and raises questions of how clinical inertia and or 'white coat' hypertension is impacting the thresholds at initiation of antihypertensive treatment. And I'd like more discussion on excluding patients when 3 or more medications are the initial prescription. I assume those are patients with their SBP at the high end of the distribution. With two classes of hypertension medications and combination medications as initial treatment being in European guidelines for hypertension treatment for some time, the author should assess including this group of patients, or at least indicate the numbers of those patients excluded.
--

REVIEWER	Joshua Barzilay Kaiser Permanente Georgia 3650 Steve Reynolds Blvd Duluth, GA 30096 USA
REVIEW RETURNED	18-May-2020

GENERAL COMMENTS	This paper examines trends at what level elevated BP is treated. I have several questions:  1. I do not understand Table 1. Perhaps it is better to say I do not understand the structure of the study. What are the number of participants in the source data? Did you include more and more participants each year? were prior participants included in each successive year? In 2007 there were nearly 11,000 participants and there were only 326 people with diabetes initiated on treatment? Is that not low? As you can see - I do not understand the study structure..... Also you have only 4% with albuminuria. This is quite low for a group with a mean age of around 60 years (my assumption of age based on the data). Generally the prevalence is higher 2. Did the study end in 2013? Why do you stop at 2013? After the ACCORD studies one assumes there would more impetus to lower BP. 3. Could you comment on trends today, especially with use of electronic medical records and the use of algorithms to prompt the primary care physician. Do you think your results at this pint are rleevant?
--

VERSION 1 – AUTHOR RESPONSE

Reviewer: 1

Reviewer Name: Gregory Wozniak

Institution and Country:

American Medical Association

United States

Please state any competing interests or state 'None declared': None declared

Please leave your comments for the authors below

Does the data include other comorbidities that could be included in the model? The authors use the number of chronic condition medications as a control, but there may be chronic conditions where no medications are prescribed. Also, the severity of those conditions and the date of hypertension diagnosis should be accounted for. Duration of diabetes is accounted for. Or are these accounted for in the frailty index. A brief description of the frailty index would be helpful for readers who are unfamiliar with the index.

#We understand the comment and agree with the reviewer's concern that comorbidities may play a role in antihypertensive treatment initiation. In the age analysis, several comorbidities were included as confounders. Based on the availability of data and on our knowledge, we included the comorbidities which we believe could affect the physician's decision when to initiate antihypertensive treatment, such as albuminuria and past cardiovascular events. Unfortunately, since only the presence or absence of a diagnosis code is available in our database, we did not have data on the

severity of those conditions. Also, the date of hypertension might be relevant but this is not documented for almost half of the patients in our database. Most of the patients that do have a date of diagnosis, were diagnosed in the year before initiation of antihypertensive medication. We now address these issues in the limitations section (page 11).

In the frailty analysis, we have compared patients with different frailty based on an index that consists of a long list of diagnoses. Many other conditions, including those where medication is not prescribed, are indeed included in this analysis. This frailty index contains 36 deficits where patients get a point for each deficit they have. The sum of the deficits divided by 36 is the indication of frailty and can take a value between 0 (patient has no deficits) and 1 (patient has all possible deficits). Each listed deficit is defined by one or several diagnoses, and therefore the index includes in total 140 ICPC codes for symptoms, diagnoses and the presence of polypharmacy (defined as at least five different chronically prescribed medications). For chronic conditions, such as chronic obstructive pulmonary disease, a diagnosis anytime in the past is considered, whereas for other short-term or episodic conditions, such as depression, only data from the past year are considered. We have extended the description of the frailty index in the methods section of the manuscript to address this comment (page 5). #

The frailty index has a range of 0-1, and was categorized in to tertiles. The tables have the ranges of 0 – 0.06, 0.08 – 0.11 and 0.14-.036. I don't see where the authors clarify only presenting those ranges, and how what is presented corresponds to tertiles of the index.

#We recognize that some unclarity regarding the use of the frailty index occurred. The tertiles used in our analysis were based on the range of frailty index values we observed in our patients. The index can in theory take values between 0 to 1. However, in reality, values in the higher range are almost impossible. The observed range in our study – now extended with data from 2014 - was between 0 and 0.40. Therefore, the tertiles were formed based on this range. Since the distribution was skewed to the right, the tertile limits are in the lower range of the values. We clarified the use of tertiles and ranges in the manuscript (page 5). #

Regarding references, just one suggestion. There are more recent guidelines for management of blood cholesterol which the authors should review, ACC/AHA 2018 Guideline on the Management of Blood Cholesterol.

#We are aware of the newer guidelines and appreciate this suggestion. However, our study focuses on the changes in recommendations in the period between 2007 and 2014. We therefore refer to the guidelines that were applicable in the time of the study period. We have adapted the introduction to clarify our focus on the study period (page 3). #

Comments on sensitivity analysis:

The authors examined the use of the mean of SBP. What about sensitivity of results where rather than using SBP within 120 days of medication initiation, some shorter look back, eg, 30, 60 or 90.? Shorter look backs may reduce the sample, but 120 days would be from a fairly old visit, and raises questions of how clinical inertia and or 'white coat' hypertension is impacting the thresholds at initiation of antihypertensive treatment.

#We agree that an SBP measurement 120 days before treatment initiation is a relatively old measurement but in some cases there is a delay between the measurement that can be conducted by a practice assistant and the decision to initiate medication by the GP. Since we used the SBP level closest to the date of initiation, most of the measurements were done less than 120 days before initiation. In the 90, 60 and 30 days before initiation, 97%, 88% and 79% of these measurements were done, respectively. Given your comment, we have conducted an additional sensitivity analysis using only the measurements in the last 30 days, which showed similar results (see below). We decided not to include this in the manuscript since delayed initiation is part of what happens in practice, and we do

not want to introduce selection bias by excluding these patients.

Table: Influence of calendar year and age or frailty on blood pressure thresholds (multilevel analysis)

β 95% CI P

AGE†

Calendar year 0.158 -1.293, 1.609 0.831 <0.001§

Calendar year² -0.126 -0.278, 0.026 0.105

Age <60 years -7.426 -10.046, -4.806 0.000

Age 60 – 69 years -3.601 -6.131, -1.072 0.005

Age 70 – 79 years -0.975 -3.512, 1.562 0.451

Age ≥80 years reference group

Interaction year*age none are significant

FRAILITY‡

Calendar year 0.530 -0.938, 1.998 0.479 <0.001§

Calendar year² -0.170 -0.324, -0.015 0.032

Frailty 0 – 0.03 -0.241 -2.098, 1.615 0.799

Frailty 0.06 – 0.09 -0.220 -2.072, 1.632 0.816

Frailty 0.11 – 0.40 reference group

Interaction year*frailty none are significant

† the age model was adjusted for sex, duration of diabetes, number of chronic medication at initiation, number and/or type of glucose lowering therapy, lipid lowering therapy, presence of albuminuria, presence of dyslipidaemia, haemoglobin A1C, history of cardiovascular events, estimated glomerular filtration rate and BMI; ‡ the frailty model was adjusted for sex, duration of diabetes, number and/or type of glucose lowering therapy, lipid lowering therapy and HbA1c

§ joint significance of calendar year and calendar year² using Wald test#

And I'd like more discussion on excluding patients when 3 or more medications are the initial prescription. I assume those are patients with their SBP at the high end of the distribution. With two classes of hypertension medications and combination medications as initial treatment being in European guidelines for hypertension treatment for some time, the author should assess including this group of patients, or at least indicate the numbers of those patients excluded.

#This is a relevant issue. We decided on this exclusion criterion based on the guidelines in the study period, which state that 'The combination of two antihypertensive drugs may offer advantages also for treatment initiation, particularly in patients at high cardiovascular risk in which early BP control may be desirable.' (European guidelines on hypertension management 2009). This recommendation is also observed in the Dutch guidelines, where a combination of two antihypertensive drugs is suggested in patients with very high SBP >180 mmHg. It is, however, not recommended to initiate treatment using three different antihypertensives. Given your comment we have looked at the number of patients that were excluded due to the initiation with three or more antihypertensives, which was 4.7% of the study population. We assume that these are patients with prevalent antihypertensive treatment, who entered the dynamic cohort during the study period. We have added a sentence to the method section to clarify this assumption (page 4). #

Reviewer: 2

Reviewer Name: Joshua Barzilay

Institution and Country:

Kaiser Permanente Georgia

3650 Steve Reynolds Blvd

Duluth, GA 30096

USA

Please state any competing interests or state 'None declared': none

Please leave your comments for the authors below

This paper examines trends at what level elevated BP is treated.

I have several questions:

1. I do not understand Table 1. Perhaps it is better to say I do not understand the structure of the study. What are the number of participants in the source data? Did you include more and more participants each year?

#The numbers in Table 1 presented in brackets after each calendar year represent the total number of patients in each calendar year that were included in the GIANTT database. Since GIANTT is a dynamic cohort, this number of patients differs per calendar year with new patients entering the cohort when either entering the general practice or being diagnosed with diabetes, or leaving it, by either leaving the practice or dying. This total number increases over the years since more general practices and more patients were included in the database. We have adjusted Table 1 of the manuscript to make this clearer (page 8) and added a flow chart explaining the exclusion of patients based on the specific exclusion criteria per year in the supplementary material (Supplementary Figure 1).#

Were prior participants included in each successive year? In 2007 there were nearly 11,000 participants and there were only 326 people with diabetes initiated on treatment? Is that not low?

#In each calendar, we included patients when they initiated antihypertensive treatment, defined as a first prescription for an antihypertensive drug with no prescription for an antihypertensive in the 365 days before the date of initiation. In theory, some patients may have been included in the analysis more than once if they discontinued antihypertensive treatment for more than 365 days and reinitiated treatment during the study period. Since our aim is to assess at what SBP levels general practitioners decide to initiate treatment, we believe this does not pose a problem. The low percentage of patients initiating antihypertensive treatment per year can be explained by the factor that this is a cohort including many prevalent patients with type 2 diabetes. The majority of the patients have had diabetes for more than 2 years (73%) and many of them are already treated with antihypertensives. It has been shown previously that 75% and 76% of patients included in the GIANTT database in 2008 and 2012, respectively, were treated with antihypertensive medication. (Sidorenkov et al, PLoS ONE 2013 8(10): e78821 and Smits et al., Diabetes Care 2017 Jul;40:e83-e84) In addition, we had to exclude some patients for various reasons. We hope that our added flow-chart clarifies these numbers (see previous point).#

As you can see - I do not understand the study structure..... Also you have only 4% with albuminuria. This is quite low for a group with a mean age of around 60 years (my assumption of age based on the data). Generally the prevalence is higher

#Again, this can be explained by the fact that we look at patients that initiate with antihypertensive treatment. In the whole population included in GIANTT, the prevalence of microalbuminuria or macroalbuminuria is much higher – more than 20%, with 78.4% of those with prevalent albuminuria already treated with a renin angiotensin aldosterone inhibitor (Hellemons et al, Nephrol Dial Transplant (2013) 28: 706–715). This illustrates that most of the patients with albuminuria in the database are already being treated with antihypertensive medication. At the time when they start antihypertensive treatment, they have a much lower chance of having elevated albuminuria levels. #

2. Did the study end in 2013? Why do you stop at 2013? After the ACCORD studies one assumes there would more impetus to lower BP.

#Our primary interest was to evaluate to what extent recommendations for more personalized treatment had been implemented in the period 2007-2013, when there were changes towards

recommending more personalized SBP targets, particularly for older and frail patients. It would indeed be of interest to continue the period of observation, and we now have extended our analysis to include 2014. Unfortunately, validated data from later years are currently not available for this patient cohort.#

3. Could you comment on trends today, especially with use of electronic medical records and the use of algorithms to prompt the primary care physician. Do you think your results at this pint are relevant?

#In the Netherlands, electronic medical records have been used for a long time and they include protocols and general prompts to help primary care physicians in providing structured diabetes care as recommended in the guidelines. Nevertheless, we observed a lack of differentiation between patients of different age or frailty after more personalized targets were recommended during our study period, that is, up to three years after changed recommendations were published. This is a major concern. Without having more recent data, we can only speculate on the trends today. However, with the additional year 2014, we do not see that the more frail patients initiate antihypertensive treatment at higher SBP levels than the less frail patients. On the contrary, the downward trend continues in the most frail group. We also do not see an indication that the youngest patients initiate treatment at lower SBP levels. Where the SBP levels at initiation for the older patients seem to decrease, the SBP level in the <60 years appears to increase in 2014. Our results illustrate that implementing changes in treatment recommendations may take considerable time. This is an important learning point for policy makers to speed up the uptake of recommendations in the future and raise awareness among physicians. Whether more specific algorithms could prompt physicians in becoming more pro-active is an area requiring further research. Algorithms based, for instance, on START criteria can be helpful to stimulate timely initiation of treatment in younger patients. Algorithms based on STOPP criteria, however, focus on patients already treated. Our study shows that more efforts are particularly needed to prevent initiation of too much treatment in older or frail patients. Such patient-specific algorithms are not yet widely implemented. We have added more reflection on the implications of our findings to the conclusions (page 13 and 14).#

We are looking forward for your feedback on our adaptations.

VERSION 2 – REVIEW

REVIEWER	Gregory D Wozniak American Medical Association USA
REVIEW RETURNED	13-Jul-2020

GENERAL COMMENTS	The authors have responded to the reviewers comments. The current paper addresses an important issue in antihypertensive medications.
---

REVIEWER	Joshua Barzilay Emory University Atlanta, Georgia USA
REVIEW RETURNED	25-Jun-2020

GENERAL COMMENTS	Just a few more small comments ABSTRACT, Results - add "grades" (or the like) after "different frailty" INTRO - remove "their" from first line INTRO line 49 - "That risk factor control has improved over time." You mean blood pressure control, not risk factor control
--

	RESULTS Variation between practices - you write ICC=0.032; in the Table it is 0.037. A very large factor that may have lowered BP levels during the study period were the electronic reminders and linking physician compensation to achieving BP targets. Thus it is hard to derive strong conclusions about what is driving lower BP levels vis a vis age and frailty. I would strongly emphasize this point
--	--

VERSION 2 – AUTHOR RESPONSE

We appreciate the opportunity for this minor revision of our manuscript and are thankful for the comments and suggestions of the reviewers. We have adapted the manuscript based on the comments from the Reviewer 2:

- *ABSTRACT, Results - add "grades" (or the like) after "different frailty"*

We have added "groups" after "different frailty" in the results section of the Abstract

- *INTRO - remove "their" from first line*

We have removed "their" from the first line in the introduction.

- *INTRO line 49 - "That risk factor control has improved over time." You mean blood pressure control, not risk factor control*

We have changed "risk factor" into "blood pressure".

- *RESULTS Variation between practices - you write ICC=0.032; in the Table it is 0.037.*

Thank you for your thorough check. We have corrected the ICC-value in Table 2 into '0.032'.

- *A very large factor that may have lowered BP levels during the study period were the electronic reminders and linking physician compensation to achieving BP targets. Thus it is hard to derive strong conclusions about what is driving lower BP levels vis a vis age and frailty. I would strongly emphasize this point*

We have adapted the conclusion with more attention for the introduction of the performance indicators.

Finally, the key points for strengths and limitations have been revised based on comments of our co-authors.

The changes that we have made are highlighted with track-changes in the manuscript.

We are looking forward for your feedback on our adaptations.